# Robust Federated Learning Through Representation Matching and Adaptive Hyper-parameters

## Abstract

Federated learning is a distributed, privacy-aware learning scenario which trains a single model on data belonging to several clients. Each client trains a local model on its data and the local models are then aggregated by a central party. Current federated learning methods struggle in cases with heterogeneous client-side data distributions which can quickly lead to divergent local models and a collapse in performance. Careful hyper-parameter tuning is particularly important in these cases but traditional automated hyper-parameter tuning methods would require several training trials which is often impractical in a federated learning setting. We describe a two-pronged solution to the issues of robustness and hyper-parameter tuning in federated learning settings. We propose a novel representation matching scheme that reduces the divergence of local models by ensuring the feature representations in the global (aggregate) model can be derived from the locally learned representations. We also propose an online hyper-parameter tuning scheme which uses an online version of the REINFORCE algorithm to find a hyper-parameter distribution that maximizes the expected improvements in training loss. We show on several benchmarks that our two-part scheme of local representation matching and global adaptive hyper-parameters significantly improves performance and training robustness.[1]

## 1 Introduction

The size of the data used to train machine learning models is steadily increasing, and the privacy concerns associated with storing and managing this data are becoming more pressing. Offloading model training to the data owners is an attractive solution to address both scalability and privacy concerns. This gives rise to the Federated Learning (FL) setting (McMahan et al., 2016) where several clients collaboratively train a model without disclosing their data. In synchronous FL, training proceeds in rounds where at the beginning of each round, a central party sends the latest version of the model to the clients. The clients (or a subset of them) train the received model on their local datasets and then communicate the resulting models to the central party at the end of the round. The central party aggregates the client models (typically by averaging them) to obtain the new version of the model which it then communicates to the clients in the next round.

The FL setting poses a unique set of challenges compared to standard stochastic gradient descent (SGD) learning on a monolithic dataset. In real-world settings, data from each client may be drawn from different distributions, and this heterogeneity can lead the local learning processes to diverge from each other, harming the convergence rate and final performance of the aggregate model. We address this challenge in two ways: (1) adaptive on-line tuning of hyper-parameters; and (2) adding a regularization term that punishes divergent representation learning across clients.

Traditional hyper-parameter tuning schemes, such as random search (Bergstra & Bengio, 2012) or Bayesian methods (Bergstra et al., 2011; Snoek et al., 2012), require several training runs to evaluate the fitness of different hyper-parameters. This is impractical in the FL setting, which strives to minimize unnecessary communication. To address this issue, we formulate the hyper-parameter

---

[1]Code used to run all experiments is available under the anonymous repository `https://gitlab.com/anon.iclr2020/robust_federated_learning`

selection problem as an online reinforcement learning (RL) problem: in each round, the learners perform an action by selecting particular hyper-parameter values, and at the end of the round get a reward which is the relative reduction in training loss. We then update the hyper-parameter selection policy online to maximize the rewards.

Unlike centralized SGD, where there is a single trajectory of parameter updates, the FL setting has many local parameter trajectories, one per each active client. Even though they share the same initial point, these trajectories could significantly diverge. Averaging the endpoints of these divergent trajectories at the end of a training round would then result in a poor model (Zhao et al., 2018). Clients with heterogeneous data distributions exacerbate this effect as each client could quickly learn representations specific to its local dataset. We introduce a scheme that mitigates the problem of divergent representations: each client uses the representations it learns to reconstruct the representations in the initial model it receives. A client is thus discouraged from learning representations that are too specific and that discard information about the globally learned representations. We show that this representation matching scheme significantly improves robustness and accuracy in the presence of heterogeneous client-side data distributions.

We evaluate the performance of our two-part scheme, representation matching and online hyper-parameter adjustments, on image classification tasks: MNIST, CIFAR10; and on a keyword spotting (KWS) task. We show that for homogeneous client-side data distributions, our scheme consistently improves accuracy. For heterogeneous clients, in addition to improving accuracy, our scheme improves training robustness and stops catastrophic training failures without having to manually tune hyper-parameters for each task.

## 2   RELATED WORK

The simplest automated hyper-parameter tuning methods execute random searches in hyper-parameter space to find the best-performing hyper-parameters (Bergstra & Bengio, 2012). Random search is outperformed by Bayesian methods (Bergstra et al., 2011; Snoek et al., 2012) that use the performance of previously selected hyper-parameters to inform the choice of new hyper-parameter to try. In these methods, the fitness of a hyper-parameter choice is the post-training validation accuracy. Running the training process to completion in order to evaluate the hyper-parameter fitness is computationally intensive. This motivates running partial training runs instead (Klein et al., 2016; Li et al., 2016). However, in FL settings with tight communication budgets, running several (partial) training runs to select the right hyper-parameters could still be impractical.

Hyper-parameters can be optimized using gradient descent to minimize the final validation loss. To avoid storing the entire training trajectory in order to evaluate the hyper-gradients, Maclaurin et al. (2015) uses reversible learning dynamics to allow the entire training trajectory to be reconstructed from the final model. Alternatively, the hyper-gradients can be calculated in a forward manner, albeit at an increased memory overhead (Franceschi et al., 2017). Using the hyper-gradient to optimize hyper-parameters is an iterative process that requires several standard training runs (hyper-iterations), again making this approach impractical in a communication-constrained FL setting. Hyper-parameters can be optimized using SGD in tandem with model parameters through the use of hyper-networks (Ha et al., 2016; Lorraine & Duvenaud, 2018) which map hyper-parameters to optimal model parameters. It is unclear how hyper-networks can be learned in a FL setting as we are interested in choosing hyper-parameters that result in the best *post-aggregation* model and not the hyper-parameters chosen by any particular client to optimize its own model on its own dataset.

The hyper-parameter tuning approaches most related to ours are those based on RL methods. Daniel et al. (2016); Jomaa et al. (2019) learn policies and/or state-action values that are then used to select good hyper-parameters. However, several training runs are needed to learn the policies and the action values. Perhaps the closest approach to ours is the method in Xu et al. (2017) which trains an actor-critic network in tandem with the main model and uses it to select actions (hyper-parameter choices) that maximize the expected reduction in training loss. It is unclear how this approach can be applied in a FL setting as the training process on each client has no access to the loss of interest which is the loss of the post-aggregation model.

FL typically struggles when the client-side data distributions are significantly different (McMahan et al., 2016; Zhao et al., 2018) as this causes the parameter trajectories to significantly diverge

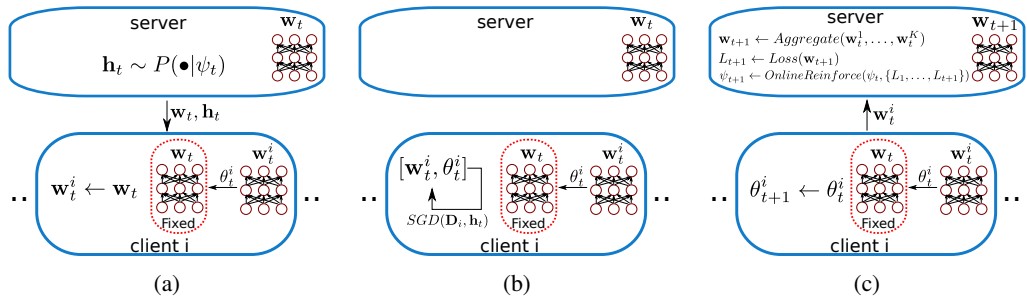

Figure 1: The three steps in a federated learning round. (a) The server communicates the latest model parameters $\mathbf{w}_t$ and the training hyper-parameters $\mathbf{h}_t$ to each client. (b) client $i$ trains the model parameters $\mathbf{w}_t^i$ and the representation matching parameters $\theta_t^i$ to minimize the classification loss and the matching loss. (c) Each client sends back the updated model parameters to the server. The server aggregates the received parameters, evaluates the loss of the aggregate model, and updates the parameters of the hyper-parameter distribution, $\psi$, based on the history of the aggregate losses.

across different clients. This severely degrades the performance of the averaged model (Zhao et al., 2018). One straightforward solution is to reduce the SGD steps each client takes between the synchronization (model averaging) points. More frequent model synchronization, however, increases communication volume. Various compression methods (Sattler et al., 2019; Konečný et al., 2016) are able to compress the model updates, which makes it possible to synchronize more frequently and mitigate the effect of heterogeneous data distributions. Sattler et al. (2019) takes the extreme case of synchronizing after every SGD step, which reduces the FL setting to standard SGD. An alternative solution is to mix the datasets of the different clients to obtain more homogeneous client-side data distributions (Zhao et al., 2018). This, however, compromises the privacy of the clients' data.

## 3 METHODS

We build upon the federated averaging (FedAvg) algorithm (McMahan et al., 2016) and introduce two novel aspects to it: global adaptive hyper-parameters and local (per-client) representation matching. We first give an informal description of the complete algorithm. We consider a synchronous FedAvg setting with $K$ clients, where $\mathbf{D}_k$ is the dataset local to client $k$. Training proceeds in rounds. Figure 1 illustrates the steps involved in training round $t$. The server maintains a distribution $P(\mathcal{H}|\psi)$ over the space of hyper-parameters $\mathcal{H}$. At the beginning of round $t$ (Fig. 1a), the selected clients receive the most recent model parameters from the server, $\mathbf{w}_t$, together with training hyper-parameters $\mathbf{h}_t$ sampled from $P(\mathcal{H}|\psi_t)$. Client $i$ uses two copies of $\mathbf{w}_t$ to initialize two models: a fixed model $\mathcal{M}^F$ and a trainable model $\mathcal{M}_t^i$. The parameters $\mathbf{w}_t^i$ of $\mathcal{M}_t^i$ are trained using SGD (Fig. 1b). Client $i$ maintains a set of local parameters $\theta_t^i$ that it uses to map the activations in $\mathcal{M}_t^i$ to the activations in $\mathcal{M}^F$. $\theta_t^i$ and $\mathbf{w}_t^i$ are simultaneously trained to minimize a two-component loss: the first component is the standard training loss of $\mathcal{M}_t^i$ on $\mathbf{D}_i$ (for example, the cross-entropy loss); the second component is the mean squared difference between the activations in $\mathcal{M}^F$ and the activations reconstructed from $\mathcal{M}_t^i$ using $\theta_t^i$. We denote this second component as the representation matching loss. In the final step in the round (Fig. 1c), each participating client sends its final model parameters, $\mathbf{w}_t^i$, to the central server. The server aggregates the model parameters, and uses the loss of the resulting model to update the parameters $\psi$ of the hyper-parameter distribution, $P(\mathcal{H}|\psi)$.

### 3.1 REPRESENTATION MATCHING

We now describe the representation matching scheme. For an example model, Fig. 2 illustrates how the local parameters $\theta^i$ in client $i$ are used to map the activations of the model being trained, $\mathcal{M}^i$, to the activations of the fixed model $\mathcal{M}^F$. $\mathcal{M}^i$ is parameterized by $\mathbf{w}^i$ while $\mathcal{M}^F$ is parameterized by $\mathbf{w}$ which was received from the server at the beginning of training. To simplify notation, we dropped the index of the round, $t$. Given a data point $\mathbf{x} \sim \mathbf{D}_i$, $\mathbf{x}$ is fed to $\mathcal{M}^i$ and $\mathcal{M}^F$ to obtain the layer activations $[\mathbf{a}_1(x; \mathbf{w}^i), \ldots \mathbf{a}_M(x; \mathbf{w}^i)]$ and $[\mathbf{a}_1(x; \mathbf{w}), \ldots, \mathbf{a}_M(x; \mathbf{w})]$, respectively. These are not all the model activations, but only the $M$ activations we are interested in matching. The

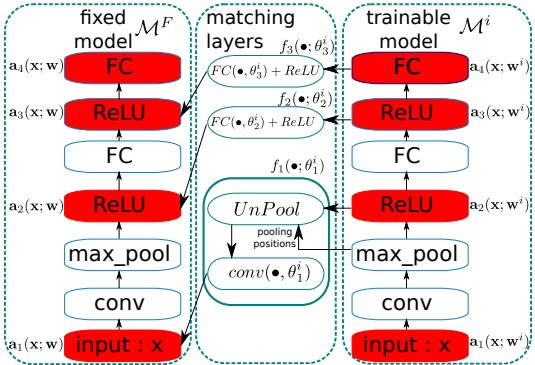

Figure 2: Illustration of representation matching using a model with one convolutional layer and two fully connected layers. Layers whose activations are used in the matching loss are shown in red. The matching layers map the activations in $\mathcal{M}^i$ to the activations in $\mathcal{M}^F$.

activations in the two models are matched by a set of matching layers parameterized by $\theta^i$ ($\theta^i \equiv [\theta^i_1, \theta^i_2, \theta^i_3]$ in Fig. 2). Instead of deriving the activations of one layer in $\mathcal{M}^F$ from the activations of the corresponding layer in $\mathcal{M}^i$, we found performance significantly improves if the activations of one layer are instead derived from the activations of the next layer of interest above it. As shown in the representative example in Fig. 2, the activations/layers of interest are the input layer, the point-wise non-linearity layers and the top layer. We use convolutional matching layers if the input has spatial information (such as within a convolutional stack), and fully-connected matching layers otherwise. To reverse a pooling operation, we follow the scheme in Zhao et al. (2015) and unpool pooled activations using the pooling positions used during the pooling operation as shown in Fig. 2.

Given a training point and label $(\mathbf{x}, y) \sim \mathbf{D}_i$, we define the matching loss at client $i$ as:

$$\mathcal{L}^i_M(\mathbf{x}; \mathbf{w}^i, \mathbf{w}, \theta^i) = \sum_{j=1}^{j=M-1} ||f_j(\mathbf{a}_{j+1}(x; \mathbf{w}^i); \theta^i_j) - \mathbf{a}_j(x; \mathbf{w})||_2^2 \tag{1}$$

where $f_j$ denotes the $j^{th}$ matching layer. The per-point training loss at client $i$ is:

$$\mathcal{L}^i(\mathbf{x}, y; \mathbf{w}^i, \mathbf{w}, \theta^i) = \mathcal{L}_C(\mathbf{a}_M(\mathbf{x}; \mathbf{w}^i); y) + \mathcal{L}^i_M(\mathbf{x}; \mathbf{w}^i, \mathbf{w}, \theta^i) \tag{2}$$

where $\mathcal{L}_C$ is the cross-entropy loss and $\mathbf{a}_M(\mathbf{x}; \mathbf{w}^i)$ is the top layer activation in the trainable model.

Our representation matching architecture is similar in some respects to auto-encoding architectures, particularly ladder networks (Rasmus et al., 2015). Unlike ladder networks, however, we do not reconstruct the activations of a non-noisy model from a noisy model but rather, reconstruct the activations of the most recent aggregate model from the activations of the model while it is being trained on a local dataset. This has a regularizing influence as it stops the clients from learning representations that are too specific to their local datasets. This should also ameliorate the effect of model divergence in the clients as all client models have to learn representations that are mappable to a common set of representations defined by the common aggregate model. Additional regularization comes about because a layer's activations are used to reconstruct the activations of a layer below it. A similar technique was used in what-where autoencoders (Zhao et al., 2015), and was shown to significantly improve accuracy. Our approach is fundamentally distinct, however, as we use the activations in one one model, $\mathcal{M}^i$, to reconstruct the activations in a different model, $\mathcal{M}^F$.

Our representation matching scheme is similar in spirit to the techniques used to learn invariant feature representations for domain adaptation where the goal is to minimize the discrepancy between the distributions of features extracted from different domains. However, current techniques for learning domain-invariant features(Ganin et al., 2016; Shen et al., 2017) are incompatible with the FL setting as they require the learner to have access to data from different domains (clients in our case) in order to match the feature distributions across these domains.

### 3.2 Adaptive hyper-parameters

At each round, the server provides the clients with training hyper-parameters. The hyper-parameters $\mathbf{h}_t$ provided in round $t$ are a sample from $P(\mathcal{H}|\psi_t)$. Let $L_t$ be the loss of the aggregate model at the beginning of round $t$ on a representative set of data points (we describe in the next subsection practical schemes for evaluating $L_t$). We define the *reward* received for choosing hyper-parameter $\mathbf{h}_t$ as $r_t = (L_t - L_{t+1})/L_t$. The use of relative loss reduction is motivated by the desire to have the scale of rewards unchanged throughout training. At round $t$, the goal is to maximize:

$$J_t = \mathbb{E}_{P(\mathbf{h}_t|\psi_t)}[r_t] \qquad (3)$$

Taking the derivative of $J_t$ and approximating it with a one-sample Monte Carlo estimate, we obtain

$$\nabla_{\psi_t} J_t = \mathbb{E}_{P(\mathbf{h}_t|\psi_t)}[r_t \nabla_{\psi_t} log(P(\mathbf{h}_t|\psi_t))] \qquad (4)$$

$$\approx r_t \nabla_{\psi_t} log(P(\mathbf{h}_t|\psi_t)) \quad where \quad \mathbf{h}_t \sim P(\mathbf{h}_t|\psi_t) \qquad (5)$$

The score-function or REINFORCE gradient estimator (Williams, 1992) in Eq. 5 can be readily evaluated and can be used to update $\psi_t$. However, the variance of this gradient estimator can be quite high (Rezende et al., 2014), especially since we are using a single-sample estimate. To reduce variance, we introduce a reward baseline (Greensmith et al., 2004) which is the weighted average reward in an interval $[t - Z, t + Z]$ centered around $t$. The update equation for $\psi_t$ is:

$$\psi_{t+1} \leftarrow \psi_t - \eta_H (r_t - \bar{r}_t) \nabla_{\psi_t} log(P(\mathbf{h}_t|\psi_t)) \quad where \quad \bar{r}_t = \gamma_Z \sum_{\substack{\tau=t-Z \\ \tau \neq t}}^{\tau=t+Z} (Z + 1 - |\tau - t|) r_\tau \quad (6)$$

where $\eta_H$ is the hyper learning rate and $\gamma_Z$ a normalizing constant. We weigh nearby rewards more heavily than distant rewards when calculating the baseline in round $t$. A causal version of Eq. 6 that only depends on past rewards is:

$$\psi_{t+1} \leftarrow \psi_t - \eta_H \sum_{\tau=t-Z}^{\tau=t} (r_\tau - \hat{r}_t) \nabla_{\psi_\tau} log(P(\mathbf{h}_\tau|\psi_\tau)) \quad where \quad \hat{r}_t = \frac{1}{Z+1} \sum_{\tau=t-Z}^{\tau=t} r_\tau \qquad (7)$$

which is the update equation for $\psi$ that we use. Even though we are using RL terminology, there are several fundamental differences between our problem and traditional RL problems. First, the setting is non-stationary as the same action in different rounds could lead to a different reward distribution. For example, large learning rates are appropriate towards the beginning of training but would increase the loss towards the end of training. In our scheme, this is reflected in the design of the baseline which weighs nearby rewards more heavily as these provide a more accurate baseline of the rewards at the current state of the learning problem. This non-stationary behavior can be modeled as a partially-observable Markov decision process (Jaakkola et al., 1995). While several techniques can be used to learn policies in non-stationary settings (Padakandla et al., 2019; Abdallah & Kaisers, 2016), these methods are not online and require several trajectories (training runs in our case) to optimize the policy. The second difference from traditional RL methods is that we do not seek to maximize the (discounted) sum of rewards, but rather, at each round $t$, we seek to maximize the rewards in a small interval $[t - Z, t]$ which is what allows us to formulate an online algorithm.

### 3.3 Full Algorithm and Practical Considerations

Algorithm 1 describes the FedAvg algorithm with our two contributions: representation matching and adaptive hyper-parameters. There are a couple of practical considerations that we address here:

**Evaluating the loss of the aggregate model:** After each round, the server needs to evaluate the loss of the aggregate model in order to obtain the reward signal (Line 15 in algorithm 1). This could be done in two ways:1) The server maintains a small validation set on which it evaluates the loss. This validation set has to be representative of the clients' data. Unlike the scheme in Zhao et al. (2015), this validation set is not shared with the clients and can be collected from them in a secure way to avoid revealing the origin of each data point. 2) At the start of a round, the clients themselves evaluate the loss of the aggregate model on a small part of their training data and then send the scalar loss to the server. The server averages these losses to obtain an estimate of the loss. This estimate is good if the fraction of participating clients in the round is high, which would make the evaluated loss more representative of the loss across all clients. In our experiments, we use the first approach.

---

**Algorithm 1** Federated averaging with representation matching and online hyper-parameter tuning. Setting with $T$ training rounds, $K$ clients, $n_k$ datapoints per client, $N$ total datapoints, and a fraction $C$ of clients participating in each round

---

1: initialize $\mathbf{w}_1$ and $\theta_1^1, \ldots, \theta_1^K$ and $\psi_1$
2: $L_1 \leftarrow Loss(\mathbf{w}_1)$
3: **for** $t$=1 to $T$ **do**
4:      $\mathbf{h}_t \sim P(\bullet|\psi_t)$                            ▷ Sample a set of hyper-parameters
5:      $S_t \leftarrow$(random selection of $CK$ clients)
6:      **for** $i \in S_t$ **do**                       ▷ Run in parallel in each client in $S_t$
7:          $\mathbf{w}_t^i \leftarrow \mathbf{w}_t$                  ▷ Receive model parameters from server
8:          **for** $n$=1 to $n\_iterations(\mathbf{h}_t)$ **do**     ▷ Number of SGD iterations as defined in $\mathbf{h}_t$
9:              $(\mathbf{X}, Y) \leftarrow sample(\mathbf{D}_i)$          ▷ Sample a training mini-batch
10:             $[\mathbf{w}_t^i, \theta_t^i] \leftarrow [\mathbf{w}_t^i, \theta_t^i] - \eta(\mathbf{h}_t)\nabla_{[\mathbf{w}_t^i, \theta_t^i]}\mathcal{L}_i(\mathbf{X}, Y; \mathbf{w}_t^i, \mathbf{w}_t, \theta_t^i)$    ▷ $\mathcal{L}_i$ as defined in Eq. 2
11:          **end for**
12:          $\theta_{t+1}^i \leftarrow \theta_t^i$
13:      **end for**
14:      $\mathbf{w}_{t+1} \leftarrow \mathbf{w}_t + \sum\limits_{i \in S_t} \frac{n_k}{N}(\mathbf{w}_t^i - \mathbf{w}_t)$
15:      $L_{t+1} \leftarrow Loss(\mathbf{w}_{t+1})$
16:      $r_t \leftarrow (L_t - L_{t+1})/L_t$
17:      $Z' \leftarrow min(Z, t-1)$
18:      $\psi_{t+1} \leftarrow \psi_t - \eta_H \sum\limits_{\tau=t-Z'}^{\tau=t} (r_\tau - \hat{r}_t)\nabla_{\psi_\tau} log(P(\mathbf{h}_\tau|\psi_\tau))$    $where$    $\hat{r}_t = \frac{1}{Z'+1}\sum\limits_{\tau=t-Z'}^{\tau=t} r_\tau$
19: **end for**

---

**Choosing the form of** $P(\mathcal{H}|\psi)$   : For $D$ hyper-parameters, we use a discrete D-dimensional hyper-parameter space. For the $p^{th}$ hyper-parameter, we have a finite set of allowable values $\mathcal{H}_p$. $\mathcal{H}$ is a D-dimensional grid containing all possible combinations of the allowed values of the $D$ hyper-parameter. $|\mathcal{H}| = \prod_{p=1}^{D} |\mathcal{H}_p|$. For $P(\mathcal{H}|\psi)$ we use a D-dimensional discrete Gaussian. Let $\mathcal{N}(\bullet|\mu, \mathbf{A})$ be a standard(continuous) D-dimensional Gaussian with mean $\mu$ and precision $\mathbf{A}$. Let $\mathbf{h}_j = (h_j^1, \ldots, h_j^D)$ be a point on the grid $\mathcal{H}$. $P(\mathbf{h}_j|\psi) = \frac{1}{\mathbf{Z}}\mathcal{N}(\mathbf{h}_j|\mu, \mathbf{A})$ where $\psi = \{\mu, \mathbf{A}\}$ and $\mathbf{Z} = \sum_{\mathbf{h} \in \mathcal{H}} \mathcal{N}(\mathbf{h}|\mu, \mathbf{A})$. Since different hyper-parameters can have different scales, we shift and normalize the allowed values for each hyper-parameter so that they have zero mean and are in the range $[-0.5, 0.5]$ before constructing the grid. The grid of hyper-parameters $\mathcal{H}$ thus has zero-mean and the same scale in all dimensions. This increase the stability of the hyper-parameter tuning algorithm as the optimization space is uniform in all directions.

We choose a discrete hyper-parameter space $\mathcal{H}$ to force different hyper-parameter choices to be significantly different, which would provide more distinct reward signals to the online REINFORCE algorithm. The choice of a uni-modal distribution, such as the discrete Gaussian, encourages hyper-parameter exploration to focus on areas around the mode. This embodies the inductive bias that the optimal hyper-parameter in one round is close to the optimal hyper-parameter in the previous round.

## 4   EXPERIMENTAL RESULTS

We evaluate the performance of our two-part scheme on the MNIST and CIFAR10 image classification datasets, and on a KWS task using the speech commands dataset (Warden, 2018). We restrict the KWS task to only ten keywords. In all experiments, we use ten clients and use one of two data distributions: **iid** where the dataset is split randomly across the ten clients, and **non-iid** where each client only has data points belonging to one of the ten classes. We use our online hyper-parameter tuning scheme to tune the learning rate and the number of SGD iterations (see algorithm 1). In all experiments, we use the same hyper-hyper-parameters controlling the hyper-parameter tuning process which are the hyper-learning rate $\eta_H$, the hyper-parameter grid $\mathcal{H}$, and the REINFORCE baseline interval $Z$. One exception is the KWS task where we modify the grid $\mathcal{H}$ to reduce the number of allowed SGD iterations per round to reflect the smaller size of the dataset. A batch size of 64 is used throughout. In all experiments, we add an entropy regularization (ER) term (Pereyra et al., 2017) to

the client losses during training. For client $i$ in round $t$ with input $\mathbf{x}$, the ER loss term has the form:

$$\mathcal{L}_{ER}(\mathbf{x}, \mathbf{w}_t^i) = max\left(0, H_{min} - H(softmax(\mathbf{a}_M(\mathbf{x}; \mathbf{w}_t^i)))\right) \tag{8}$$

where $H(\bullet)$ is the entropy operator and $\mathbf{a}_M$ the top layer activity. $\mathcal{L}_{ER}$ penalizes highly confident (low entropy) output distributions when their entropy falls below $H_{min}$. This loss term significantly improves the performance of the FedAvg algorithm in the presence of non-iid client data.

As a comparison baseline, we run experiments with a fixed decay schedule for both the learning rate and the number of SGD iterations per round. As a baseline for our representation matching scheme, we run experiments where each client's training loss is augmented with a term that penalizes weight divergence between the client model parameters and the initial parameters received from the server. For client $i$ in round $t$, this weight divergence (WD) loss term has the form $||\mathbf{w}_t - \mathbf{w}_t^i||_2^2$. This penalty function was introduced into the FL setting by the FedProx algorithm (Li et al., 2018) to mitigate the effect of model divergence.

We report results for the standard FedAvg algorithm (FA), FedAvg with a weight divergence loss term in the clients (FA+WD), and FedAvg augmented with adaptive hyper-parameters(AH) and/or representation matching (RM). We consider two values for $C$ (the fraction of clients participating in each round): $C = 0.5$ and $C = 1.0$. For the baselines, FA and FA+WD, we manually tuned the hyper-parameters decay schedule to obtain best accuracy. This tuning was done separately for the **iid** and **non-iid** cases . All experiments were repeated 3 times.

**MNIST** : We use a fully-connected network with two hidden layers with 100 neurons each. As shown in table 1, the best performing algorithm is FA+AH. The use of representation matching causes a slight degradation in performance (FA+RM+AH and FA+RM). It is interesting to look at the trajectory of the mean $\mu$ of the discrete Gaussian hyper-parameter distribution $P(\mathcal{H}|\mu, \mathbf{A})$. This is shown in the first column of Fig. 3 for the **iid** and **non-iid** cases. In the **non-iid** case, the online REINFORCE algorithm pushes the mean learning rate down as it detects that a smaller learning rate yields greater loss reductions, while in the **iid** case, the mean learning rate is pushed up instead. This aligns with what a practitioner would do to ensure convergence in the more challenging **non-iid** case. The REINFORCE algorithm chooses to keep the mean number of SGD iterations per round relatively high for most of the training run. As shown in table 1, these choices yield slightly better accuracy (FA+AH) than the fixed training schedule used in the standard FA algorithm.

Table 1: MNIST accuracy figures

| | Data Distribution | FA | FA+WD | FA+RM+AH | FA+RM | FA+AH |
|---|---|---|---|---|---|---|
| C=1.0 | iid | $97.9 \pm 0.1$ | $97.9 \pm 0.005$ | $97.9 \pm 0.06$ | $97.8 \pm 0.06$ | $\mathbf{98.0 \pm 0.04}$ |
| | non-iid | $94.7 \pm 0.07$ | $94.8 \pm 0.2$ | $94.7 \pm 0.08$ | $94.5 \pm 0.3$ | $\mathbf{95.5 \pm 0.04}$ |
| C=0.5 | iid | $97.4 \pm 0.06$ | $97.3 \pm 0.03$ | $98.1 \pm 0.06$ | $97.4 \pm 0.05$ | $\mathbf{98.2 \pm 0.01}$ |
| | non-iid | $92.1 \pm 0.8$ | $92.0 \pm 0.4$ | $92.5 \pm 0.1$ | $91.6 \pm 0.3$ | $\mathbf{93.2 \pm 0.8}$ |

**CIFAR10** : We use a network with two convolutional layers (with 32 and 64 feature maps and 5x5 kernels) followed by two fully connected layers (with 1024 and 10 neurons). 2x2 max pooling was used after each convolutional layer. As shown in table 2, the use of representation matching yields a significant improvement in accuracy for the **non-iid** case while slightly improving performance in the **iid** case. The benefit of adaptive hyper-parameters over a fixed hyper-parameter schedule are more equivocal. The evolution of adaptive hyper-parameters is shown in the second column of Fig. 3. Unlike the MNIST case, the REINFORCE algorithm chooses to push down the learning rate for the **iid** case as well which actually leads to better performance in the **iid** case compared to the fixed schedule (FA vs. FA+AH and FA+RM+AH vs. FA+RM in table 2). In the **non-iid** case, the fixed schedule performs better, though.

**Keyword spotting task** : We calculate the mel spectrogram for each speech command to obtain a 32x32 input to our network. The network we use has four convolutional layers with 3x3 kernels and 64 feature maps each (with 2x2 max pooling after each pair), followed by two fully connected layers of 1024 and 10 neurons. In this deeper network, representation matching is essential in

Table 2: CIFAR10 accuracy figures

|  | Data Distribution | FA | FA+WD | FA+RM+AH | FA+RM | FA+AH |
|---|---|---|---|---|---|---|
| C=1.0 | iid | $81.9 \pm 0.3$ | $81.4 \pm 0.2$ | $\mathbf{84.3 \pm 0.1}$ | $83.5 \pm 0.1$ | $83.4 \pm 1.2$ |
|  | non-iid | $44.2 \pm 0.9$ | $44.3 \pm 0.8$ | $52.4 \pm 2.2$ | $\mathbf{52.9 \pm 0.05}$ | $44.8 \pm 5.8$ |
| C=0.5 | iid | $76.8 \pm 0.3$ | $76.6 \pm 0.2$ | $\mathbf{79.2 \pm 1.9}$ | $76.0 \pm 0.2$ | $82.0 \pm 1.04$ |
|  | non-iid | $33.4 \pm 1.0$ | $34.4 \pm 0.9$ | $39.8 \pm 3.8$ | $\mathbf{43.3 \pm 0.8}$ | $19.8 \pm 7.0$ |

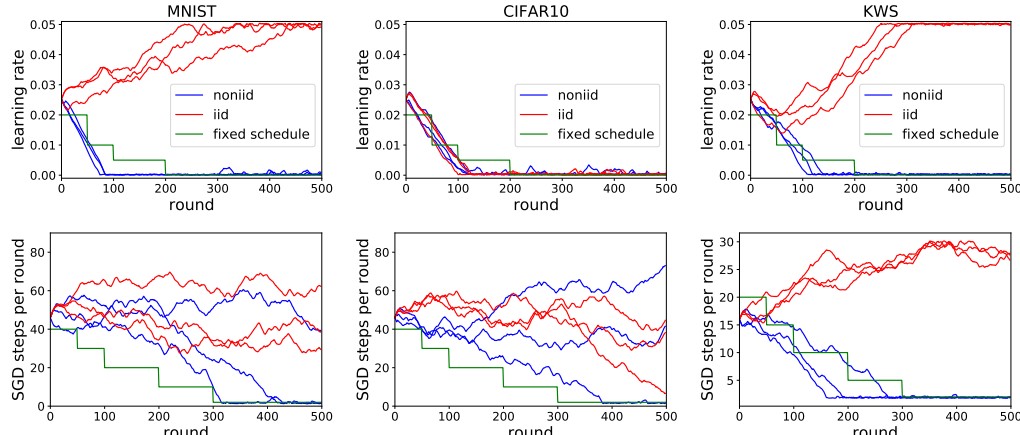

Figure 3: Evolution of the mean of the hyper-parameter distribution $P(\mathcal{H}|\psi)$ for the iid and non-iid cases. Results taken from the FA+RM+AH algorithm when $C = 1.0$. The evolution of the means are shown separately for the learning rate (first row) and for the number of SGD steps per round (second row), with one column each for the MNIST, CIFAR10, and keyword spotting (KWS) tasks. During training, the means are forced to stay within the ranges defined by the hyper-parameter grid.

the **non-iid** case as shown in table 3 since training consistently fails in its absence. It might be possible that with fine tuning of hyper-parameters, training would be possible in the **non-iid** case. However, representation matching obviates the need for such fine tuning. FA+RM+AH consistently outperforms all other methods indicating the hyper-parameters schedule found by the REINFORCE algorithm (third column in Fig. 3) is better than the fixed schedule.

Table 3: Keyword spotting task accuracy figures

|  | Data Distribution | FA | FA+WD | FA+RM+AH | FA+RM | FA+AH |
|---|---|---|---|---|---|---|
| C=1.0 | iid | $93.0 \pm 0.2$ | $93.5 \pm 0.2$ | $\mathbf{94.4 \pm 0.2}$ | $92.9 \pm 0.3$ | $94.0 \pm 0.4$ |
|  | non-iid | $28.4 \pm 7.2$ | $29.9 \pm 7.3$ | $\mathbf{81.1 \pm 0.5}$ | $79.2 \pm 0.6$ | $9.8 \pm 0.2$ |
| C=0.5 | iid | $91.4 \pm 0.3$ | $93.0 \pm 0.4$ | $\mathbf{94.3 \pm 0.1}$ | $91.0 \pm 0.2$ | $93.2 \pm 0.4$ |
|  | non-iid | $11.45 \pm 1.7$ | $11.8 \pm 2.7$ | $\mathbf{74.9 \pm 2.5}$ | $60.7 \pm 3.3$ | $10.0 \pm 0.3$ |

**Communication and computational overhead**   : Our representation matching scheme does not introduce any communication overhead between the clients and the central server. The adaptive hyper-parameters scheme introduces a negligible communication overhead for sending two scalar hyper-parameters to the clients each round. As for computational overhead, we quantify the wall-clock run-time[1] for training a client for 30 SGD iterations or mini-batches (mini-batch size of 64). Where adaptive hyper-parameters are used, we also include the time needed to execute the adaptive

---

[1]Experiments were run on a machine with a single TitanXP GPU, 32G of RAM, and an Intel Xeon E5-2699A processor.

hyper-parameter tuning procedure. The results are shown in table 4. We note that the overhead of our hyper-parameter tuning procedure is negligible (FA vs. FA+AH) and is typically less than $2\%$. This negligible computational overhead is primarily the cost of a single inference pass to evaluate the loss $L_t$ on a very small subset of the training set. The representation matching scheme is more demanding as as it requires optimizing a more complicated loss during each SGD iteration at the client.

Table 4: Wall-clock run-time in seconds. Mean and std. from 200 trials.

| Task | FA | FA+WD | FA+RM+AH | FA+RM | FA+AH |
|---|---|---|---|---|---|
| MNIST | $0.39 \pm 0.007$ | $0.41 \pm 0.008$ | $0.43 \pm 0.007$ | $0.42 \pm 0.007$ | $0.39 \pm 0.007$ |
| CIFAR10 | $0.50 \pm 0.02$ | $0.54 \pm 0.01$ | $0.67 \pm 0.02$ | $0.66 \pm 0.02$ | $0.51 \pm 0.02$ |
| KWS | $1.83 \pm 0.05$ | $1.96 \pm 0.05$ | $2.03 \pm 0.04$ | $2.01 \pm 0.04$ | $1.85 \pm 0.05$ |

## 5 DISCUSSION

We described two additions to the FedAvg algorithm: representation matching and adaptive hyper-parameters. We compared representation matching (FA+RM) against a scheme based on penalizing weight divergence (FA+WD) and FA+RM came out on top by a significant margin in the more difficult tasks we tried: CIFAR10 and KWS. We compared our adaptive hyper-parameter scheme against a fixed hyper-parameter schedule chosen based on experience. In most cases, adaptive hyper-parameters outperform the fixed-schedule scheme. The combination of our two new additions (FA+RM+AH) slightly improves performance compared to standard FA in the small MNIST network and significantly improves performance on the deeper networks used in CIFAR10 and KWS.

The hyper-parameter schedule chosen by our REINFORCE algorithm exhibits some easily interpretable behavior such as reducing the learning rate in the more difficult **non-iid** cases. However, we see some behavior that does not have an immediately obvious interpretation. For example, in order to maximize rewards in the **non-iid** case, reducing the learning rate is more important than reducing the number of SGD iterations per round as can be seen in Fig. 3. It is important to note that at each round, our online REINFORCE algorithm only seeks to maximize the loss improvement in the last few rounds. Moreover, our algorithm continuously samples hyper-parameters around the mean to determine the direction of highest reward. These noisy hyper-parameter choices could prove problematic in settings that depend on highly tuned schedules and which have no room for online exploration of hyper-parameters. We thus expect our scheme would be outperformed by traditional automated hyper-parameter tuning methods that tune hyper-parameters to maximize the final validation loss. However, our scheme is significantly more efficient as it does not require repeated training and introduces only a small computational overhead in each round.

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
