# OpenReview forum: "Robust Federated Learning Through Representation Matching and Adaptive Hyper-parameters"
_ICLR.cc/2020/Conference — Reject_

### Official Review · AnonReviewer3 · 2019-10-21
**Official Blind Review #3**

**Rating:** 3

**Review:**

- Good paper. However, the theoretical novelty is quite limited. There is no guarantee whatsoever whether the good empirical results achieved on the three experimented would persist with other datasets. Similarly, there is no analysis of the conditions required so that such empirical superiority would hold.

- The flow of the ideas in the paper is clear and unequivocal. However, writing has a lot of room for improvement in terms of typos and grammatical mistakes.
Examples
 -- p1: "are become"
 -- p5: "several fundamental difference"

- I think an analysis and comparisons based on computational run-time would be necessary to check how the RL-based methodology would fare w.r.t. FA and the other FL frameworks in comparison.

- What about comparing to other federated learning frameworks, i.e. other than versions and extensions of FA?

- In the Introduction, it was promised that the proposed method would stop catastrophic training failures; has that been actually described in the experiments?

- Similarly so for the robustness issue (which is also the first word in the paper title), where is the empirical demonstration of the robustness of the proposed method?

- Regarding the last paragraph in Section 2.2 and the first fundamental difference between the proposed modelling choice and RL, is the former still different from non-stationary RL? An example of recent works on non-stationary RL is "Reinforcement learning in non-stationary environments" by Padakandla et al. 2019.

- This is not strictly necessary, but might be a done in a future work or so: Did you consider comparing to methods which are based on learning invariant representations (which is quite similar to the methodology pursued in the paper) adopted in different ML paradigms like domain adaptation?


**Experience Assessment:**

I have read many papers in this area.

**Review Assessment: Checking Correctness Of Derivations And Theory:**

I carefully checked the derivations and theory.

**Review Assessment: Checking Correctness Of Experiments:**

I carefully checked the experiments.

**Review Assessment: Thoroughness In Paper Reading:**

I read the paper thoroughly.

---

> ### Author Response · Authors · 2019-11-08
> **Response**
>
> We thank the reviewer for the useful comments.
> - We acknowledge that we can not theoretically prove that our method will always perform better than the baselines. That is why we demonstrated the superiority empirically. Like the vast majority of machine learning advances (for example: word2vec, transformers, resnets, DQN), the novelty in our case lies in the formulation of the architecture and the loss functions, and the way we optimize these loss functions. We have empirically shown improved performance in several benchmarks, and tried to provide clear intuitive explanation for why that should be the case (the representation matching loss encourages local models to learn representations that are mappable to the global model representation, and hence reduce overfitting on the local dataset. Optimizing the REINFORCE loss moves the hyper-parameters towards regions with higher rewards or higher loss improvements per round).
>
> - We apologize for the typos. These have been fixed in the revised edition.
>
> - We now describe the communication and computational overhead of our method in an additional paragraph in the results section and added a table with the run-times of all methods. The computational overhead of our adaptive hyper-parameter scheme is negligible (< 2%), while the overhead of our representation matching scheme is more noticeable (~32% in the worst case)
>
> - We tried to compare against the strongest baselines we could find that still respect the constraints of the federated learning (FL) setting. Work that deals with FL in non-iid settings that we are aware of often violate these constraints. For example, [Zhao et al. 2018] shares data between clients to improve data homogeneity which compromises data privacy (one of the core pillars of FL). [Sattler et al. 2019] synchronize the client models after every SGD step, making it more akin to distributed SGD than FL. We took the standard FA algorithm and strengthened it significantly in the non-iid case through the use of entropy regularization (we are the first to use entropy regularization to mitigate the effects of model divergence in FA). In the revised paper, we have manually tuned the hyper-parameters for these baselines to further improve their accuracy, but they still fall significantly short of the accuracy of our method in more complex benchmarks (cifar10 and KWS).
>
> - Stopping catastrophic training failure has been demonstrated for the non-iid case in the keyword spotting task (table 3) . Our baselines (FA and FA+WD) have very poor accuracy and in some cases perform at chance level. Our method FA+RM+AH is consistently able to learn and performs at much better accuracy (50%-60% absolute accuracy improvement).
>
> - By robustness, we mean our method is robust to heterogeneity in the client-side data distributions (non-iid data case), and does not require precise tuning of hyper-parameters. This is demonstrated by its superior performance  (tables 2 and 3) compared to the baselines. For the MNIST case (table 1), the benefits of our method are not significant as the dataset is simple and the network is very shallow.
>
> - The hyper-parameter tuning process can indeed be considered as a non-stationary RL problem. We now mention this at the end of section 3.2. We added the sentence "This non-stationary behavior can be modeled as a partially-observable Markov decision process [1]. While several techniques can be used to learn policies in non-stationary settings [2][3], these methods are not online and require several trajectories (training runs in our case) to optimize the policy"
>
> - Yes. our representation matching scheme is similar in spirit to the techniques used to learn invariant feature representations for domain adaptation where the goal is to minimize the discrepancy between the distributions of features extracted from different domains. It is certainly interesting to look at the methods used to minimize cross-domain feature discrepancies in domain adaptation, and see if they can be adapted to the FL setting. Currently the domain adaptation methods we are aware of require the learner to have access to data from the different domains (in order to match the feature distributions). This is incompatible with the FL setting where learning is done locally in each client (domain) with no access to data from other clients (domains). Thank you for bringing this connection to our attention. We added a sentence to the end of section 3.1 to highlight this connection.
>
> [1]Jaakkola, et al. "Reinforcement learning algorithm for partially observable Markov decision problems." Advances in neural information processing systems. 1995.
> [2]Padakandla, Sindhu, and Shalabh Bhatnagar. "Reinforcement Learning in Non-Stationary Environments." arXiv preprint arXiv:1905.03970 (2019).
> [3]Abdallah, Sherief, and Michael Kaisers. "Addressing environment non-stationarity by repeating Q-learning updates." The Journal of Machine Learning Research 17.1 (2016): 1582-1612.

---

### Official Review · AnonReviewer1 · 2019-10-23
**Official Blind Review #1**

**Rating:** 3

**Review:**

In this paper, the authors propose a novel representation matching scheme to reduce the divergence of local models in federated learning. In addition, the authors propose an online hyper-parameter tuning scheme. The paper is well-written. The empirical results show good performance. In overall, I think the authors propose an interesting alternative of weight regularization (also called weight divergence loss in this paper).

Detailed comments:

1. (Very minor, does not affect the score) The baseline FA+WD is actually the same as FedProx proposed in [1].

2. (Major concern) In most experiments, there is a huge gap in the performance between FA+WD and FA+RM. However, it is unclear such improvement is caused by the better robustness of RM, or simply simply caused by bad hyperparameters of FA+WD. Since FA, FA+WD, and FA+RM have different loss functions, it is unreasonable and unfair to use the same hyperparameters for them. The authors should report results with fine-tuned hyperparameters, so that we can confirm that RM really works. Otherwise, the results of the experiments are questionable.

3. It seems that AH is irrelevant to federated learning. Even if we use fully synchronous SGD to train the model, we can still use AH to tune the hyperparameter on the server side. Ah does have some contribution, but seemingly it doesn't really contribute to the federated learning algorithm.



----------------
Reference

[1]  Li, Tian et al. “Federated Optimization for Heterogeneous Networks.” (2018).

**Experience Assessment:**

I have published one or two papers in this area.

**Review Assessment: Checking Correctness Of Derivations And Theory:**

N/A

**Review Assessment: Checking Correctness Of Experiments:**

I assessed the sensibility of the experiments.

**Review Assessment: Thoroughness In Paper Reading:**

I read the paper at least twice and used my best judgement in assessing the paper.

---

> ### Author Response · Authors · 2019-11-08
> **Response and baseline hyper-parameter tuning**
>
> We thank the reviewer for the useful feedback and comments.
> 1)Thank you for pointing this out. Indeed FA+WD is the same as FedProx. We refer to this work in the revision and we added the proper citation.
>
> 2)We  did tune the weight divergence coefficient in FA+WD to obtain best results. For methods without our adaptive hyper-parameter scheme, we did not tune the learning rate schedule or the number of SGD iterations per round schedule. We believe such tuning is unrealistic in a federated learning setting where you typically do not have the luxury to run multiple training trials to pick the best hyper-parameter schedules. However, to address your concern, we now fine-tune the hyper-parameter schedules for FA and FA+WD for the iid and non-iid cases. Fine-tuning these schedules in the iid case did not lead to an appreciable improvement in accuracy. Fine-tuning for the non-iid case, however, improved accuracy significantly in the cifar10 and keyword spotting task. We now report these improved accuracies in the paper. These accuracies still fall significantly short of FA+RM even though the hyper-parameter schedule for FA+RM  was not tuned. We would like to stress that such tuning in the FA and FA+WD case is not realistic as the optimal tuned hyper-parameter schedules in the iid and non-iid cases are different. Thus, in order to obtain these new improved results for FA and FA+WD, the federation has to know beforehand how data is distributed on the clients. The fact that FA+RM still outperforms such a strong but unrealistic baseline speaks well for its robustness. We have updated the code repository to use the new tuned schedules for FA and FA+WD for the CIFAR10 and KWS tasks.
>
> 3)We agree that our adaptive hyper-parameters scheme (AH) can also be used in standard synchronous SGD. The fact that AH is applicable in learning scenarios other than federated learning does not imply that it is irrelevant to FL. Indeed, we believe the generality of AH is a strength, not a weakness. AH is a novel and general method for efficient online tuning of hyper-parameters that does not require multiple training runs. This makes it eminently suitable for the FL setting with its tight computation and communication constraints. AH performs well in comparison to a reasonable hyper-parameter schedule chosen based on experience. We believe this makes it a valuable tool, especially in FL settings with unfamiliar datasets for which a human practitioner is unable to draw on experience in order to choose an appropriate hyper-parameter schedule.

---

### Official Review · AnonReviewer2 · 2019-10-23
**Official Blind Review #2**

**Rating:** 6

**Review:**

This manuscript proposes two strategies to improve both the robustness and accuracy of local agents under the setting of federated learning. Specifically, online reinforcement learning is used to perform adaptive hyperparameter search in order to maximize the utility of local models. This is quite an interesting idea since traditional hyperparameter tuning techniques, including random search or Bayesian optimization, usually requires access to monolithic dataset, which is clearly impractical under federated learning. The second contribution is an idea on using local distribution matching in order to synchronize the learning trajectories of different local models. This again is a novel and interesting idea. Overall the paper is well-written and clear to follow, which is a plus. Detailed comments and questions follow:

-   I understand that change of hyperparameter in local model affects the global model during in model aggregation stage. However, under the federated learning setting, if the number of local models is huge, then the influence of a single local model should be small. Hence instead of using online reinforcement learning for hyperparameter search, which is notoriously data-inefficient, why not framing the problem as a pure online learning problem? This helps to increase the data efficiency and also increases the stability of learning.

-   In the design of matching network, is there any intuition why performance significantly improves if the activations of
one layer are instead derived from the activations of the next layer of interest above it? Furthermore, why we need to have additional model with parameters $\theta_i$ for alignment? Intuitively if we want the new model $M$ to be close to the original one $M^F$, shouldn't we just use some distance measure, e.g., $\ell_2$ norm, to measure the distance of feature activations in the corresponding layers directly?

-   The experiments are illustrative, but might be too toyish under the federated learning setting. I appreciate the ablation studies the authors performed to show the relative impact of different strategies, which makes the relative contributions more clear. However, even in the case of MNIST and CIFAR, the improvement over baseline FA is not very significant. In this case, it would be better if the authors could also report the computational overhead in terms of running time.

**Experience Assessment:**

I do not know much about this area.

**Review Assessment: Checking Correctness Of Derivations And Theory:**

I assessed the sensibility of the derivations and theory.

**Review Assessment: Checking Correctness Of Experiments:**

I assessed the sensibility of the experiments.

**Review Assessment: Thoroughness In Paper Reading:**

I read the paper at least twice and used my best judgement in assessing the paper.

---

> ### Author Response · Authors · 2019-11-08
> **Authors' response**
>
> We thank the reviewer for the positive feedback. Below, we address the three questions/points raised:
>  - Indeed, RL is usually data-inefficient. However, the strength of RL methods is that they do not require the gradients of the reward (in this case, relative loss reduction) with respect to the optimization variables (in this case, training hyper-parameters). Obtaining such a gradient can be prohibitively expensive as we explain in the `related work' section, but using RL methods allows us to sidestep this issue. Moreover, our online RL scheme optimizes a very small number of parameters : 5 parameters for the mean and variance of the 2D discrete Gaussian distribution over the 'learning rate' and 'SGD iterations per round'. For such a small optimization space, the sample-inefficiency of an RL method does not hurt us too much, as even with a few action-reward samples, our online RL scheme learns to properly navigate the small optimization space (by raising the learning rate for the easy iid case, and lowering it for the more difficult non-iid case, for example).
>
>  - Forcing the activation of one layer to be derived from the activation of the layer of interest above it has a regularizing influence on the training of the local models. This regularizing influence turns out to be important in our case as overfitting of the local models on the local datasets is a primary issue in federated learning. That is because the local dataset available to each local model is typically a small subset of the entire data used to train the model. The regularizing influence of reconstructing the activation of earlier layers from the activations of deeper layers has been widely observed before (See for example [1] and [2]). In our case, we go one step further by reconstructing the activations of the global model from the activations of the local model (instead of doing the reconstruction solely within the local model). That is partly the reason why we need the matching/alignment parameters $\theta$ as two successive layers of interest do not typically have the same dimensions, so we need an intermediate set of parameters, $\theta$, to project one set of activations into the space of another, and then we take the L2 distance between the two as a cost to be minimized.
>
>  - We agree that the performance improvement of our method (adaptive hyper-parameters + representation matching) is not significant in the MNIST case. In the MNIST case, the network we used is very shallow (two hidden layers), and the dataset is simple enough that our strong baseline method (Federated averaging + entropy regularization + weight divergence regularization) is enough to stabilize training and achieve strong performance. For deeper networks and more difficult datasets like CIFAR10 and keyword spotting, however, our method significantly improves accuracy compared to the baselines (compare the 'FA+RM+AH' column in tables 2 and 3 to the 'FA+WD' column). This improvement is more pronounced in the more challenging non-iid case, and also more pronounced in the keyword-spotting task where we have used a deeper network than the network in the CIFAR10 case.
>    Regarding computational overhead, we added a new paragraph and run-time results table to the results section to discuss this overhead. The results indicate that using our adaptive hyper-parameters scheme has a negligible overhead. Thank you for bringing this point to our attention as we believe adding these run-time numbers strengthens our argument regarding the efficiency of our adaptive hyper-parameter scheme. The computational overhead of our Representation Matching scheme is more pronounced, though we believe it is still acceptable given the significant accuracy improvements in our more difficult benchmarks. It is important to keep in mind that the representation matching scheme does not introduce any communication overhead between the clients and servers. The adaptive hyper-parameters scheme also does not introduce additional communication overhead (apart from the negligible overhead of sending two scalar hyper-parameters to the clients each round).
>
>
> [1]Erhan, Dumitru, et al. "Why does unsupervised pre-training help deep learning?." Journal of Machine Learning Research 11.Feb (2010): 625-660.
> [2]Zhao, Junbo, et al. "Stacked what-where auto-encoders." arXiv preprint arXiv:1506.02351 (2015).

---

### Author Response · Authors · 2019-11-08
**Paper revision submitted**

We thank the reviewers for their comments. We submitted a revision of the paper to address the concerns raised. The most notable changes in the paper are:
1)To address the concern over the computational overhead of our method that has been raised by several reviewers,  we added a discussion of this overhead to the results section, together with a table comparing the actual run-times of the different methods. The table shows that the overhead for our adaptive hyper-parameters scheme is negligible (~2%) and that the overhead of our representation matching scheme is more significant but still small (~32% in the worst case)

2)To address the concern raised by reviewer 1, we tuned the hyper-parameter schedules for the baseline methods: FA and FA+WD to further improve the performance of the baselines. After hyper-parameter tuning, these baselines still fall far short of our representation matching method in the more difficult non-iid case in the cifar10 and keyword spotting tasks, even though we did not tune the hyper-parameters for our method.

We modified the text in several places to incorporate many of the useful comments from the reviewers. Please see our responses below.

---

### Decision · Program_Chairs · 2019-12-19

**Decision:**

Reject

**Comment:**

This manuscript proposes strategies to improve both the robustness and accuracy of federated learning. Two proposals are online reinforcement learning for adaptive hyperparameter search, and local distribution matching to synchronize the learning trajectories of different local models.

The reviewers and AC agree that the problem studied is timely and interesting, as it addresses known issues with federated learning. However, this manuscript also received quite divergent reviews, resulting from differences in opinion about the novelty and clarity of the conceptual and empirical results. Taken together, the AC's opinion is that the paper may not be ready for publication.